# Assessment of Full-Fat *Tenebrio molitor* as Feed Ingredient for *Solea senegalensis*: Effects on Growth Performance and Lipid Profile

**DOI:** 10.3390/ani14040595

**Published:** 2024-02-11

**Authors:** Ismael Hachero-Cruzado, Mónica B. Betancor, Antonio Jesús Coronel-Dominguez, Manuel Manchado, Francisco Javier Alarcón-López

**Affiliations:** 1Instituto Andaluz de Investigación y Formación Agraria, Pesquera, Alimentaria y de la Producción Ecológi-ca (IFAPA), Centro El Toruño, Junta de Andalucía, Camino Tiro Pichón s/n, 11500 El Puerto de Santa Ma-ría, Cádiz, Spain; ismael.hachero@juntadeandalucia.es (I.H.-C.); manuel.manchado@juntadeandalucia.es (M.M.); 2“Crecimiento Azul”, Centro IFAPA el Toruño, Unidad Asociada al CSIC, 141500 El Puerto de Santa María, Cádiz, Spain; 3Institute of Aquaculture, School of Natural Sciences, University of Stirling, Stirling FK9 4LA, Scotland, UK; m.b.betancor@stir.ac.uk; 4Beetle Genius 3.0, 41500 Alcalá de Guadaira, Sevilla, Spain; ajcordom@gmail.com; 5Dept Biología y Geología, Ceimar-Universidad de Almería, Ctra. de Sacramento s/n, 04120 Almería, Spain

**Keywords:** *Tenebrio molitor*, insect meal, sustainable protein sources, *Solea senegalensis*, fatty acid profile

## Abstract

**Simple Summary:**

Yellow mealworm is considered as one of the most promising protein sources for replacing fish meal in aquafeeds, among other things because is rich in protein, a good source of micronutrients and exhibits low carbon footprint and land usage. The aim of this study was to evaluate the effects of substituting partial plant or marine-derived ingredients with full-fat yellow mealworm meal at two different levels on the growth performance and fatty acid profiles of Senegalese sole. For this purpose, the study tested a control diet and four experimental diets. Two of the experimental diets substituted marine-derived ingredients for insect meal at two levels (5 and 10%), while the other two substituted components of vegetable origin for insect meal (10 and 15%). The addition of insect meal resulted in an increase in growth rate in both cases, whether replacing fish or plant meals. This led to a decrease in muscle total lipid while maintaining the relative levels of n-3 PUFA and DHA, ultimately improving the lipid health indices n-3: n-6.

**Abstract:**

*Tenebrio molitor* (TM) is considered as one of the most promising protein sources for replacing fish meal in aquafeeds, among other things because it is rich in protein, a good source of micronutrients and has a low carbon footprint and land use. However, the main drawback of TM is its fatty acid profile, in particular its low content of n-3 PUFA. This study evaluates the effects of partially replacing plant or marine-derived with full-fat TM meal at two different levels on growth performance and lipid profiles of Senegalese sole (*Solea senegalensis*). For this purpose, a control diet (CTRL) and four experimental isoproteic (53%) and isolipidic (16%) diets were formulated containing 5 and 10% TM meal replacing mostly fish meal (FM5 and FM10), or 10 and 15% TM meal replacing mostly plant meal (PP10 and PP15). Fish (215 g) were fed at 1% of their body weight for 98 days. The final body weight of fish fed the experimental diets containing TM meal was not different from that of fish fed the CTRL diet (289 g). However, the inclusion of TM meal resulted in a gradual improvement in growth rate and feed efficiency in both cases (replacement of fish or plant meals), and significant differences in specific growth rate (SGR) were observed between fish fed the CTRL diet (SGR = 0.30% day^−1^) and those fed diets with the highest TM meal content (PP15; SGR = 0.35% day^−1^). The experimental groups did not show any differences in the protein content of the muscle (19.6% *w*/*w*). However, significant differences were observed in the total lipid content of the muscle, with the FM10, PP10, and PP15 groups having the lowest muscle lipid contents (2.2% ww). These fish also showed the lowest neutral lipid content in muscle (6.6% dw), but no differences were observed in the total phospholipid content (2.6% dw). Regarding the fatty acid profile, fish fed FM10, PP10 and PP15 had lower levels of linoleic acid (18:2n-6) and higher levels of oleic acid (18:1n-9) in liver and muscle compared to fish fed CTRL. However, no differences were found between fish fed CTRL and TM-based diets for docosahexaenoic acid (22:6n-3) and total n-3 PUFA in liver and muscle. In conclusion, our study demonstrated that full-fat TM inclusion up to 15% in *S. senegalensis* diets had no negative effects or even some positive effects on fish survival, growth performance, nutrient utilization and flesh quality.

## 1. Introduction

The global demand for fish has led to a significant expansion of aquaculture, resulting in a sharp increase in prices for raw materials used in aquafeeds, especially fish meal (FM) and fish oil (FO) [1]. Moreover, aquaculture competes with other animal production systems for feed ingredients, further increasing the demand for raw materials. There is also an urgent need to find sustainable alternative ingredients for aquaculture diets due to the high environmental cost of feed [2]. Plant meals, such as soy and corn meals, are the most common alternatives to FM, becoming competitive in 2006 when the price of FM notably increased [3]. However, they have an unbalanced essential amino acid (EAA) profile, specifically lacking in methionine and lysine, and containing high levels of anti-nutritional factors (ANFs), causing pro-inflammatory effects in the fish intestine [4,5]. In addition, there has been intense competition for plant protein resources in recent years, which are currently being used for both human consumption and terrestrial animal feed. This competition has resulted in a significant price increase, limiting their use as an aquaculture alternative [1]. To promote the sustainable growth of aquaculture and to meet future feed demand, it is crucial to develop alternative protein and oil sources, such as seaweed, algae, microalgae, single-cell proteins, microbial biomass and insects, as well as to recycle food waste [6].

Insect meals (IM) are becoming increasingly popular as a primary ingredient in aquafeeds. One of the advantages of insects is their ability to grow on organic by-products and bioconvert a wide range of organic wastes, contributing to a circular economy [7]. Furthermore, they have a relatively low carbon footprint and land use [8,9]. Insects are a highly nutritious food source, rich in protein, lipids and essential amino acids [10]. Most edible insect species appear to be good sources of valuable nutrients and compounds that modulate animal microbiota, improving animal health [11]. According to research, yellow mealworm (*Tenebrio molitor*) has been identified as a highly promising protein source that could partially replace FM in aquafeeds for aquaculture production [12]. It has been found that *T. molitor* (TM) meals are rich in crude protein (53.2%) and fat (34.5%) [10,13], with an adequate amino acid profile, although limited in total sulfur amino acids methionine and cystine [10,14]. They are also an excellent source of zinc, selenium, riboflavin, biotin, pantothenic acid and folic acid [10]. However, it should be noted that they may be deficient in calcium, vitamin D3, vitamin A, vitamin B12, thiamine, vitamin E, iodine, manganese and sodium [10]. One disadvantage of TM meals is their high lipid content and the lack of n-3 long-chain polyunsaturated fatty acids (LC-PUFA), such as eicosapentaenoic acid (EPA, 20:5n-3) and docosahexaenoic acid (DHA, 22:6n-3) [15]. Although defatting TM meals is possible, careful consideration is needed as the processes for fat extraction and protein purification may reduce the environmental sustainability and profitability of IM and may affect its nutritional and functional value [16]. Therefore, understanding the effects of insect meal inclusion on the growth performance, survival and fatty acid (FA) profiles of edible fish parts is essential. 

Previous studies on Mediterranean marine species, such as European sea bass (*Dicentrarchus labrax*) and gilthead sea bream (*Sparus aurata*), have shown that full-fat TM meals can be included with up to 25% of inclusion in diets with a low terrestrial plant content (<25%) without affecting growth performance [17,18,19]. In diets for European sea bass that contain medium levels of terrestrial plant components (40%), the inclusion rates can be increased up to 80% when defatted meals are used, with no effect on growth [20]. With regard to the effects of full-fat TM meal on the FA profile of fish, previous studies have reported an increase in oleic acid (OA, 18:1n-9) and linoleic acid (LA, 18:2n-6) and a decrease in n-3 long-chain polyunsaturated fatty acids (n-3 LC-PUFAs) [17,21]. This suggests that the use of full-fat TM may have a negative effect on the lipid health index of fillets, including the n-3/n-6 ratio and atherogenicity and thrombogenicity index.

Senegalese sole (*Solea senegalensis*) is a flatfish that holds high commercial value in southern Europe and is an emerging species for marine aquaculture. In comparison to European sea bass and sea bream, which are strictly carnivorous fish with limited de novo fatty acid biosynthetic capacity, Senegalese sole is highly efficient in the biosynthesis of LC-PUFAs such as DHA [22,23]. It shows optimal survival and growth rates when fed with diets low in fish oil and reduced levels of DHA and EPA, making it an interesting model for new diets in Europe [22,23]. To date, no studies have evaluated the inclusion of TM meal in diets for Senegalese sole. The objective of this study was to evaluate the effects of replacing vegetable or marine ingredients with full-fat TM meal on the growth performance and FA profiles of Senegalese sole. To achieve this objective, Senegalese sole were fed diets containing 5% and 10% TM meal that replaced mostly fish meal (FM5 and FM10), or 10% and 15% TM meal that replaced mostly plant meal (PP10 and PP15), and a control diet (CTRL) during a 98-day growth trial. The study also assessed differences in liver and muscle lipid profiles associated with the dietary treatments.

## 2. Materials and Methods

### 2.1. Source and Composition of TM Meal

The company Beetle Genius S.L. (Brussels, Belgium) provided insect meal from *T. molitor*. Table 1 and Table 2 show the proximate and lipid compositions. The total protein and lipid content were 43.8% and 35.0%, respectively (Table 1). The lipid fraction had high levels of triacylglycerides (TAG) and free fatty acids (FFA) (Table 2). The most abundant FAs were OA (18:1n-9), 51.7% of total FAs, LA (18:2-6), 18.0% of total FAs, and palmitic acid (PA, 16:0), 18.0% of total FAs (Table 2).

### 2.2. Experimental Diets

Five iso-nitrogenous and isolipidic experimental diets were formulated, as shown in Table 1 and Table 2. Two of the diets, named FM5 and FM10, contained 5% and 10% dw TM meal, respectively, which replaced mostly fish meal. The other two experimental diets, named PP10 and PP15, included 10% and 15% dw TM meal, respectively, which replaced mostly plant meals. The fifth diet, CTRL, was TM-meal-free and used as the control. All the diets were modified to achieve the same lipid content by adjusting the oil content, mainly soy oil. They were formulated and manufactured by the Service of Experimental Diets from CEIMAR—University of Almería (Almeria, Spain), using standard aquafeed processing procedures. Briefly, all ingredients were mixed in a 120 L mixer and ground to 0.5 mm using a hammer mill (UPZ 100, Hosokawa-Alpine, Augsburg, Germany). The diets were extruded using a twin-screw extruder (Evolum 25, Clextral, Firminy, France), fitted with adequate die plates to manufacture 2 and 3 mm sinking pellets. The extruder barrel consisted of four sections and the temperature profile in each section (from inlet to outlet) was 100 °C, 95 °C, 95 °C and 90 °C, respectively. After extrusion, the pellets were dried at 30 °C in a 12 m^3^ drying chamber with forced-air circulation (Airfrio, Almería), and cooled at ambient temperature. On the following day, a vacuum oil coating was applied using a Pegasus PG-10VC LAB vacuum coater (Dinnissen, The Netherlands). The feeds were then stored in sealed plastic bags at −20 °C until use.

The formulation and chemical composition of the experimental diets are shown in Table 1 and Table 2. Crude protein and total lipids of all experimental diets were approximately 53% and 16%, respectively, on a dry-matter basis (Table 1). All diets had similar levels of total saturated FAs (Table 2), with slight differences in PA (16:0), which were proportional to the TM meal inclusion. The greatest differences between diets were related to total monounsaturated FAs (MUFAs) and n-6 and n-3 PUFAs. The control diet (CTRL) had the lowest content of MUFA, mainly OA (18:1n-9), and the highest content of n-3 LC-PUFAs, mainly DHA (22:6n-3) and EPA (20:5n-3), and n-6 PUFA, mainly LA (18:2n-6). Inclusion of TM meal increased the FFA content and decreased the TAG content.

### 2.3. Fish and Experimental Design

Senegalese sole specimens were provided by the Aquaculture company Cupimar (San Fernando, Cádiz, Spain). Fish were transported to IFAPA El Toruño (El Puerto de Santa Maria, Cadiz, Spain) and acclimated in 5000 L tanks for six months in a flow-through circuit. Two weeks prior the experiment 1050 fish were injected intraperitoneally with an electronic Passive Integrated Transponder (PIT)-tag transponder (Trovan, Fish-Tags©, Melton, UK), as previously described by Carballo, et al. [24]. The fish were then randomly distributed into fifteen tanks (five dietary treatments and three replicates per treatment), with a total volume of 360 L and a density of 70 fish per tank. The tanks were connected to a recirculation system (RAS). All tagged fish were monitored for two weeks prior to the trial and no mortality or tag losses were recorded. Subsequently, the fish were weighed and PIT-Tags were automatically registered using a FISH Reader Weight (Zeuss, Trovan, Spain). The average mean weight at the beginning of the experiment was 215 ± 5 g, with no statistically significant differences between tanks (Table 3).

During the trial, temperature, salinity and oxygen were monitored daily. The values were within the ranges of 19–24 ℃, 38–41 ppt and 3–7 ppm. The diets were supplied using automatic feeders (Mirafeed©; Innovaqua, Lebrija, Spain) in 72 doses between 01:00 and 19:00 h. The amount of feed supplied was adjusted weekly to fit the expected total biomass (based on the latest sampling, 0.5–1% of total biomass). Additionally, the tank feed supplied was adjusted to account for the daily feed left over in tanks.

All procedures were previously authorized by the Bioethics and Animal Welfare Committee of IFAPA and registered with the National authorities for regulation of animal care and experimentation under the number 22/11/2021/182. 

### 2.4. Fish Sampling

Individuals were weighed on a monthly basis for three months, starting from the beginning of the study. Sampling was performed at t0 = initial weight, and t1 = 43, t2 = 64, and t3 = 98 days. Before each sampling, fish were fasted for one day and anesthetised before handling (2- MS-222, 200 ppm). Weights and PIT-Tags were automatically recorded using a FISH Reader Weight (Zeuss, Trovan, Spain). At the last sampling, thirty fish were sacrificed with an anaesthetic overdose (MS-222, >500 mg/L). Six fish were sacrificed by diet (two fish by tank). Fish were necropsied in the cold to collect liver and muscle samples. Samples were immediately frozen at −80 °C in liquid nitrogen and stored at −80 °C until analysis.

### 2.5. Growth Performance and Nutrient Utilization

Fish growth performance was evaluated using the following formula: specific growth rate (SGR, % d^−1^) = (Ln (Wf) − Ln (Wi)/∆days) × 100, where Wf and Wi were the final and initial fish weight, respectively. Nutrient utilization indices were estimated using the following formulas: feed conversion ratio (FCR) = total feed intake on dry basis (g) × weight gain (g)^−1^, where total feed intake refers to the total amount of feed supplied; protein efficiency ratio (PER) = WG × total protein ingested (g)^−1^, where WG was the weight gain (g).

### 2.6. Biochemical Analysis

#### 2.6.1. Proximate Composition of Diets and Tissues

Gross proximate compositions of feeds (protein, lipid, ash and moisture) and fish tissues (protein and lipid) were determined according to standard procedures [25]. Briefly, moisture contents were obtained after drying the sample in an oven at 110 °C for 24 h and ash contents were determined after incineration at 600 °C for 16 h. Crude protein was measured by determining nitrogen content (N × 6.25) using automated Kjeldahl analysis (Tecator Kjeltec Auto 1030 analyser, Foss, Warrington, UK) and total lipids were extracted from feeds and tissues of the experimental fish and quantified according to the method of Folch et al. [26].

#### 2.6.2. Total Lipids, Lipid Classes and FA Analyses

For lipid extraction, approximately 200 mg of ground feed or fish tissues were placed in ice-cold chloroform/methanol (2:1, by vol) and homogenised with an Ultra-Turrax tissue disrupter (Fisher Scientific, Loughborough, UK). Next, the non-lipid and lipid layers were separated by adding 0.88% (*w*/*v*) KCl. The upper aqueous layer was then aspirated and discarded, whereas the lower organic layer was dried under oxygen-free nitrogen. The lipid content was determined gravimetrically after drying overnight in a vacuum desiccator.

The main lipid classes were separated on 20 × 10 cm plates by double development high-performance thin-layer chromatography (HPTLC) following the technics described by Olsen and Henderson [27]. Firstly, the plates were pre-run in diethyl ether and then activated at 120 °C for 1 h. The lipid classes were visualized after spraying with 3% (*w*/*v*) copper acetate containing 8% (*v*/*v*) phosphoric acid by charring at 160 °C for 20 min. Quantification was performed by densitometry using a CAMAG-3 TLC scanner (Version Firmware 1.14.16; CAMAG, Muttenz, Switzerland) with winCATS Planar Chromatography Manager. Samples and authentic standards were run alongside, in the same conditions, on high-performance thin-layer chromatography (HPTLC) plates to determine the identities of individual lipid classes by contrasting Rf values.

Fatty acid methyl esters (FAME) of total lipids were prepared by acid-catalysed transesterification at 50 °C for 16 h according to Christie [28]. Firstly, the FAME were separated and quantified by gas–liquid chromatography (Agilent Technologies 7890B GC System) using a 30 m × 0.32 mm i.d. fused silica capillary column (SUPELCOWAXTM-10, Supelco Inc., Bellefonte, PA, USA) and on-column injection at 50 °C. Hydrogen was used as carrier gas and temperature programming was from 50 °C to 150 °C at 40 °C per min and then to 230 °C at 2.0 °C per min. Then, individual methyl esters were identified by comparison with known standards and by reference to published data [29,30]. Agilent Technologies Openlab CDS Chemstation for Windows (version A.02.05.21, Santa Clara, CA, USA) was used to collect and process data.

### 2.7. Statistical Analysis

Mean and standard error of the mean (SEM) were calculated using SPSS statistics v22 software (IBM, Armonk, USA). A two-way repeated measures ANOVA was conducted to assess the effects of diets (five levels: “CTRL”, “FM5”, “FM10”, PP10” and “PP15”), and time (four levels: “t0”, “t1 = 43 days”, “t2 = 64” and “t3 = 98”; and three levels for SGR: “t0–t1”, “t1–t2” and “t2–t3”) on growth performance (fish weight and SGR). The model also included the factor “Tank”. The between-subjects factors were “Diet” and “Tank”, while “Time” was the within-subjects factor. The effects of diets on initial and final fish weights on nutrient utilization parameters (FCR and PER) and on muscle and liver protein and lipid composition were investigated using 1-way ANOVA. When ANOVA indicated a significant difference (*p* < 0.05) for a given factor, the source of the difference was identified using a Tukey test. Additionally, principal component analysis (PCA) was conducted on FA matrices to ordinate the samples.

Prior to conducting ANOVAs, normality was assessed using the Kolmogorov–Smirnov test (*p* > 0.05) and homogeneity of variances was assessed using the Levene test (*p* > 0.05). For the repeated-measures ANOVA, the Mauchly test of sphericity was conducted and sphericity corrections were applied to adjust the degrees of freedom when necessary. As proportions were compared, an arcsine transformation was performed [31]. If variances remained heterogeneous even after data transformation, untransformed data were still analysed using ANOVA, as it is a robust statistical test that is relatively unaffected by the heterogeneity of variances [32]. In such cases, the level of significance was reduced to <0.01 to avoid type I errors.

## 3. Results

### 3.1. Growth Performance and Proximate Composition

No mortality was observed through the whole trial for any of the diets. Fish growth was monitored by weight and SGR, as shown in Figure 1. Using a longitudinal approach, a statistically significant interaction diet × time (within-subjects) for weight and SGR was identified. At the beginning of the experiment, all treatments showed the lowest SGR values, probably related to adaptation to the experimental conditions. In subsequent samplings, the highest SGR was observed in the PP15 group. On average, we found a significantly higher mean SGR (between-effects) during the experimental period for PP15 compared to CTRL. Moreover, dietary inclusion of TM meal slightly improved nutrient utilisation, as determined by FCR and PER, although these differences were not statistically significant (*p* > 0.05; Table 3).

Total lipid and protein contents of fish muscle are shown in Figure 2. In general, the inclusion of TM meal in the diet reduced the total lipid content in muscle (*p* < 0.05) while maintaining protein levels (*p* > 0.05). The groups fed with a higher level of insect meal showed the lowest values of total lipids in muscle. For liver, the average total lipid content was 29% (in % wet weight, ww), with no differences between experimental groups (*p* > 0.05) (data not shown). 

### 3.2. Lipid Classes and Fatty Acid Profile

Significant differences were found in the total lipid content of muscle among the dietary treatments, and the composition of lipid classes in this tissue was analysed (Figure 3). Interestingly, no significant differences were found for polar lipids among dietary treatments (*p* < 0.05). However, a lower content of neutral lipids such as sterols, triacylglycerols and other neutral lipids (mainly free FA and sterol esters) was observed in sole fed TM-based diets. This depletion was more pronounced at the highest level of insect inclusion, regardless of the type of dietary ingredient replaced (fish or plant meals).

Changes in liver and muscle FA profile were explored by a principal component analysis (PCA) multivariate analysis. The first two components of PCA explained 86% of the total variance in liver (Figure 4A). PC1 (70.4% total variance) correlated positively with OA (r = 0.800; *p* < 0.0005) and negatively with LA (r = −0.503; *p* < 0.005). PC2 (16.0% total variance) correlated negatively with DHA (r = −0.801, *p* < 0.0005) and positively with LA (r = 0.517, *p* < 0.0025). The two-way ANOVA analyses confirmed the significant differences for FA that correlated with the PCA axis, except for DHA (Figure 4B). It should be noted that the inclusion of insect meal in diets resulted in an increase in the OA and a decrease in the LA content in the liver.

Regarding the PCA analysis for muscle, the first two PCA components explained 89.7% of the total variance (Figure 5A). PC1 (49.5% total variance) correlated positively with DHA (r = 0.648, *p* < 0.0005) and OA (r = 0.373, *p* < 0.025), and negatively with LA (r = −0.575, *p* < 0.0005); clustering the samples in two main groups according to dietary inclusion of TM meal: (i) fish fed diets CTRL and FM5 (higher content of LA); and (ii) fish fed FM10, PP10 and PP15 (higher content of OA). PC2 (40.3% total variance) correlated positively with OA (r = 0.797, *p* < 0.0005) and negatively with DHA (r = −0.545, *p* < 0.0005). The two-way ANOVA analyses confirmed the significant differences for the FA that correlated with the PCA axis, showing an increase in OA and a decrease in LA content in muscle with the inclusion of TM meal in the diet. For DHA, differences related to dietary treatments were not clearly associated with the dietary inclusion of insect meal, and no differences were found among fish fed CTRL and the diets PP10, PP15, FM5 and FM10 (Figure 5B). 

The liver and muscle FA profiles of fish fed the dietary treatments are shown in detail in Table 4 and Table 5. The most abundant FA found in liver and muscle of Senegalese sole were PA (16:0), OA (18:1n-9), LA (18:2n-6) and DHA (22:6n-3), reflecting those of the experimental diets. In liver, OA and hexadecenoic fatty acid (16:1n-9) were higher in fish fed TM-based diets. In contrast, LA, linolenic acid (LNA, 18:3n-3) and eicosatrienoic acid (20:3n-3) were higher in the liver of fish fed the CTRL diet. However, no differences were found between fish fed the CTRL and TM-based diets for eicosapentanoic acid (EPA, 20:5n-3), docosapentanoic acid (DPA, 22:5n-3), DHA, n-3 PUFA and n-3 PUFA/n-6 PUFA ratio. In muscle, the effect of insect inclusion on the FA profile was similar to that observed in liver, with higher levels of hexadecenoic acid (16:1n-9) and OA, and lower of vaccenic acid (18:1n- 7), LA (18:2n-6), LNA (18:3n-3), eicosatrienoic acid (20:3n-3), eicosatetraenoic acid (20:4n-3), and EPA detected in fish fed the TM-based diets. Meanwhile, no significant differences were found between fish fed the CTRL and TM-based diets for DHA and n-3 PUFAs. Interestingly, the higher n-3 PUFA/n-6 PUFA ratio was found in fish fed the FM10, PP10 y PP15 diets. 

## 4. Discussion

Recently, there has been an increasing interest in the use of insect meal as feed ingredient in aquaculture due to its sustainability and nutritional value [33]. However, one of the biggest drawbacks of full-fat insect meal is its fatty acid profile. In this study, we investigated the effects of the partial replacement of plant or marine-derived ingredients with full-fat TM meal on the growth performance and lipid profile of Senegalese sole.

The growth in the experimental groups was similar to that reported by other authors [34,35]. The initial values were low, possibly due to animal adaptation to the experimental conditions. However, growth recovered later. The results showed that the partial replacement of fish meal by full-fat *T. molitor* meal (TM) in sole diets (up to 10% replacement) had no negative effect on growth (final weight and SGR), nutrient utilisation (FCR and PER) and fish survival, whereas the partial replacement of plant meals by TM (up to 15% replacement) significantly improved growth rate. In general, it is difficult to compare the present results with previous reports due to species-specific characteristics and differences between the tested diets. To put our results into perspective, it is important to point out that our control group (CTRL) had a low level of total marine ingredients (46%) in order to reflect the current levels used in commercial diets. A previous study conducted with rainbow trout (*Oncorhynchus mykiss*), tench (*Tinca tinca*) and sea bream used experimental diets similar to those used in the present work in terms of levels of marine ingredients and fish meal replacements, and found no statistical differences in weight gain between control and fish fed diets containing up to 10% TM meal [36]. Instead, other works with European sea bass [17] and sea bream [19] formulated reference diets with a high levels of marine ingredients (over 64%) and both concluded that the optimal level of inclusion of full-fat TM meal in diet was 25%. Some authors have attributed the effects of insect inclusion on growth performance to the modulation of fish gut microbiota and the prebiotic activity of chitin [37,38,39]. Furthermore, the effect of chitin on protein digestibility is well established in the literature as chitin has a high protein binding capacity [19]. However, the effect of chitin in fish nutrition is dose-dependent and it is also considered to be an antinutritive compound that can reduce feed palatability, nutrient digestibility and fish growth [40]. In this work, the chitin content of TM-based diets was less than 1% dry weight (data not shown), which is below the minimum level of dietary chitin that can cause a reduction in fish growth [17,19]. To further investigate this, it would be interesting to test higher levels of TM meal in the diet, since in both cases (marine and plant ingredients replacements) we observed a trend towards improved growth performance when we increased the TM content. Nevertheless, it is worth noting that the high lipid content of full-fat TM meals may limit their inclusion in the diet as it would drastically reduce the n-3 PUFA content. This limitation would not be present if plant-based meals were substituted.

Concerning the effect of full-fat TM inclusion on the muscle and liver of Senegalese sole, it was observed that including full-fat insect meal in the diet led to a decrease in the total lipid content of the muscle, without changing the total lipid content of the liver. It is interesting to note that the reduction in muscle total lipids was due to a decrease in the major neutral muscle lipids: triacylglycerols and cholesterol. Jeong et al. [41] and Belforti et al. [42] reported a decrease in total lipids in olive flounder (*Paralichthys olivaceus*) and rainbow trout. However, other studies have reported no significant changes in the composition of fillet of blackspot sea bream (*Pagellus bogaraveo*) [43], mandarin fish (*Siniperca scherzeri*) [44], or rainbow trout [45], with the inclusion of insect meal in their diet. Differences in muscle lipid content in sole were likely related to the effect of chitin in reducing fat digestibility and lipid absorption [40,46]. The use of chitinous polymers as a feed additives in animal diets has been shown to reduce cholesterol and triacylglycerol levels in rabbits and rats without adversely affecting normal growth [47]. Due to its chemical properties, chitin can reduce the activity of digestive enzymes, as demonstrated for porcine pancreatic lipase [48]. In this sense, Hansen et al. [49] found a decrease in bile acid concentration in the pyloric gut of fish fed a diet high in chitin. The authors suggest that this reduction in bile acids may have an impact on lipase activation and efficient fatty acid absorption, especially in the pyloric region where lipid digestion mainly occurs. The growth enhancement observed in sole in this experiment could also be attributed to a reduction in lipid absorption and fat digestibility. Previous studies have shown that low dietary lipid levels can improve nutrient retention and growth in Senegalese sole, without significant effects on the whole-body composition [23]. Furthermore, it has been suggested that dietary lipid levels may have a significant effect on the expression of genes related to growth in this species [50].

The study found that fish fed TM-diets exhibited an increase in TM-associated fatty acids, including oleic acid (18:1n-9), while the levels of linoleic acid (18:2n-6), the primary component of soy oil, decreased. However, there were no significant differences in the n-3 PUFA and DHA (22:6n-3) contents between fish that were fed TM-based diets and the CTRL group. This pattern was observed in both tissues analysed: liver and muscle. Several authors have reported that including TM meal in the diet increases the content of oleic and linoleic acid while decreasing n-3 PUFA in fish tissues [17,21,41,42,44,51]. However, our observations suggest a different pattern for LA content in sole tissues. The reason for the difference in results can be attributed to two factors. Firstly, the relatively low content of LA in the TM meal used in this study (18% TFA) compared to other studies (35% TFA) [17,42,52]. Secondly, in this study, the dietary soy oil was reduced to compensate for the high lipid content of the TM meal. Furthermore, it is interesting to note that the levels of n-3 PUFA and DHA in sole muscle remained stable, despite the reduction in dietary n-3 PUFAs (with a maximum reduction of 26%) and specifically in DHA (with a maximum reduction of 23%). This result could be explained by the DHA biosynthesis and/or the selective deposition of the dietary DHA, as this FA is usually accumulated in tissues at higher concentrations than those present in the diets [53]. In terms of PUFA biosynthesis capacity, Senegalese sole has a ∆4 Fads2, which allows it to produce DHA from docosapentaenoic acid (22:5n-3) via the so-called “∆4 pathway” [54]. As a result, the replacement of marine ingredients with terrestrial sources in their diet has shown good growth performance for Senegalese sole, while maintaining the flesh content of n-3 PUFA [22,55].

## 5. Conclusions

In conclusion, this study demonstrates that replacing of up to 10% of fish meal with full-fat *T. molitor* (TM) meal in Senegalese sole diets has no negative effect on fish growth, nutrient utilisation and survival. Furthermore, replacing up to 15% of plant meals with TM significantly improves growth rate. Additionally, TM-based diets reduce neutral lipids while maintaining total muscle protein in both cases of marine and vegetable replacements. Despite the absence of n-3 PUFAs in TM meal, the study found that the relative levels of n-3 PUFAs and DHA were maintained, and the lipid health index n-3:n-6 were improved. The study confirms that the TM meal is a viable alternative to the fish meal and vegetable meal ingredients in sole diets, ensuring optimal growth and muscle quality.

## Figures and Tables

**Figure 1 animals-14-00595-f001:**
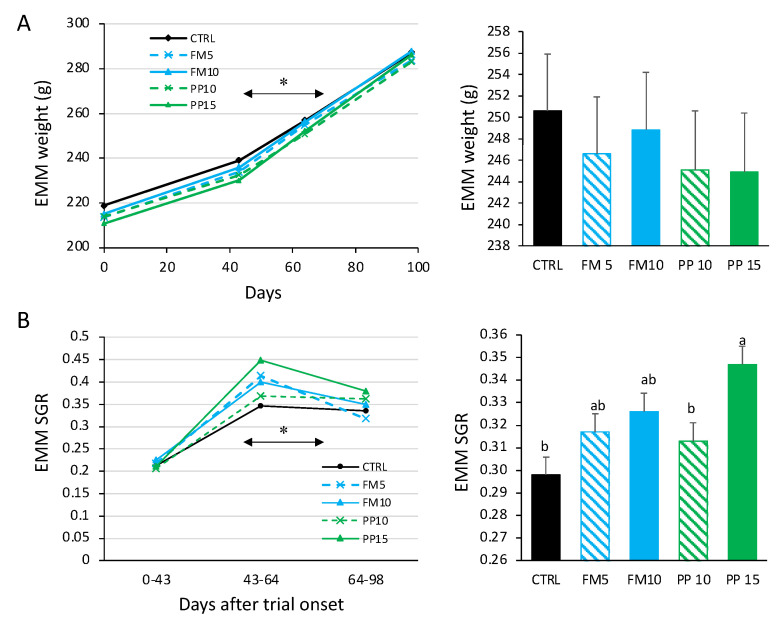
Weight (**A**) and specific growth rates (SGR) (**B**) of Senegalese sole fed with the experimental diets. Lines represent the estimated marginal means (EMM) for weight or the SGR in the evaluation periods, as calculated by repeated measures ANOVA. Asterisks on the horizontal denote significant differences for interaction diet × time (within-subjects). Bars on the right represent the EMM for weight and SGR in the whole period. Different letter denotes statistically significant differences according to the between-subjects. FM diets are in blue and PP diets in green. The lowest TM inclusion within each dietary group is indicated with dashed lines.

**Figure 2 animals-14-00595-f002:**
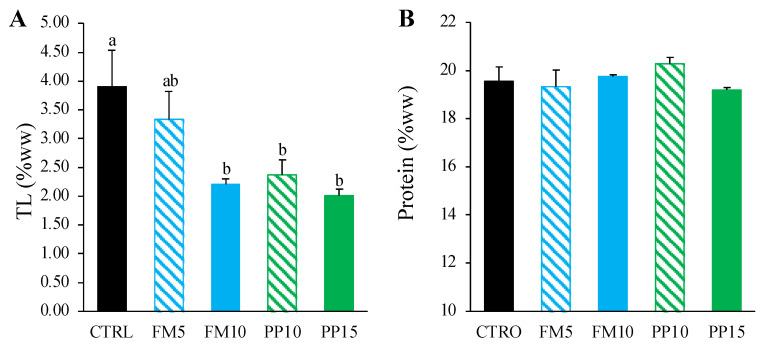
Total lipids (in % wet weight, ww) (**A**) and total protein (in % ww) (**B**) in muscle of Senegalese sole fed with the experimental diets after 98 days. Data are presented as mean ± SEM (*n* = 6). Different letters above bars indicate significant differences among dietary treatments (Tukey test; *p* < 0.05). FM diets are in blue and PP diets in green. The lowest TM inclusion within each dietary group is indicated with dashed lines.

**Figure 3 animals-14-00595-f003:**
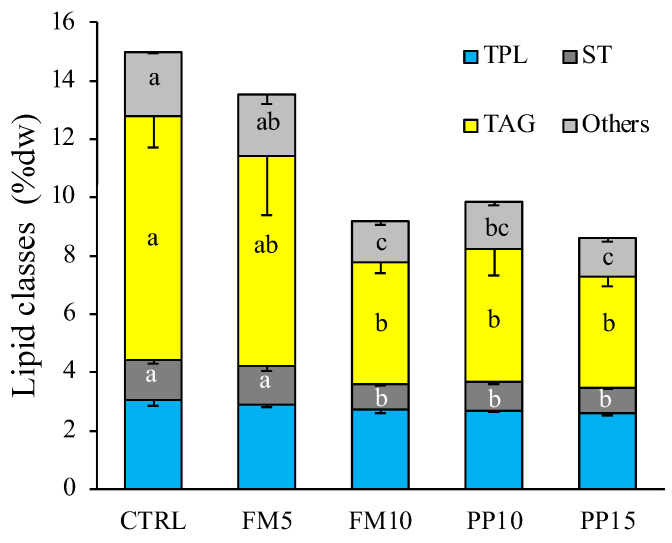
Lipid classes composition (in % dw) in muscle of sole fed with the experimental diets. Data are presented as mean ± SEM (*n* = 6). Different letters above bars indicate significant different between dietary treatments (Tukey test; *p* < 0.05). TPL, total polar lipids; ST, sterols; TAG, triacylglycerols; Others, others neutral lipids. TPL include lyso-phosphatidylcholine, sphingomyeline, phosphatidylcholine, phosphatidylserine, phosphatidylinositol, phosphatidylglycerol, phosphatidylethanolamine; others include diacylglycerols, free fatty acids and sterol esters.

**Figure 4 animals-14-00595-f004:**
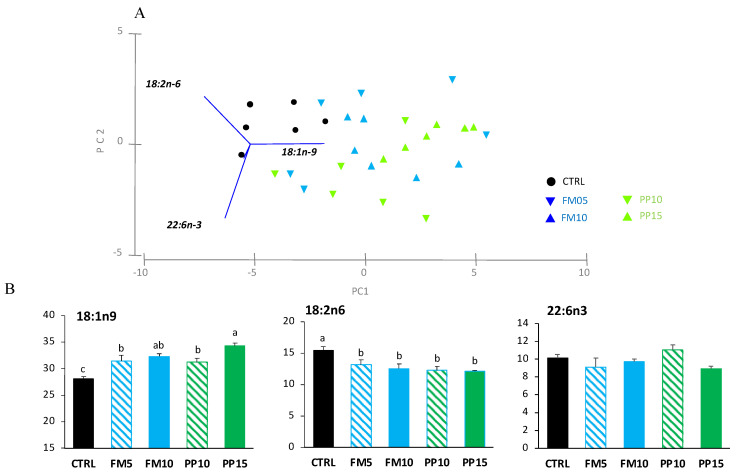
Hepatic FA content in fish fed with the control (CTRL, black) or TM-based diets replacing fish meal (blue) or replacing plant meal (green). (**A**) Principal component analysis (PCA) plot based on FA composition (% TFA); (**B**) hepatic content of FA significantly correlated with PC1: OA (18:1n-9) and LA (18:2n-6); and PC2: DHA (22:6n-3). Data are expressed as mean ± SEM (*n* = 6). Results of two-way ANOVA are presented in the square. Different letters above bars indicate significant different between dietary treatments (Tukey test; *p* < 0.05). FM diets are in blue and PP diets in green. The lowest TM inclusion within each dietary group is indicated with dashed lines.

**Figure 5 animals-14-00595-f005:**
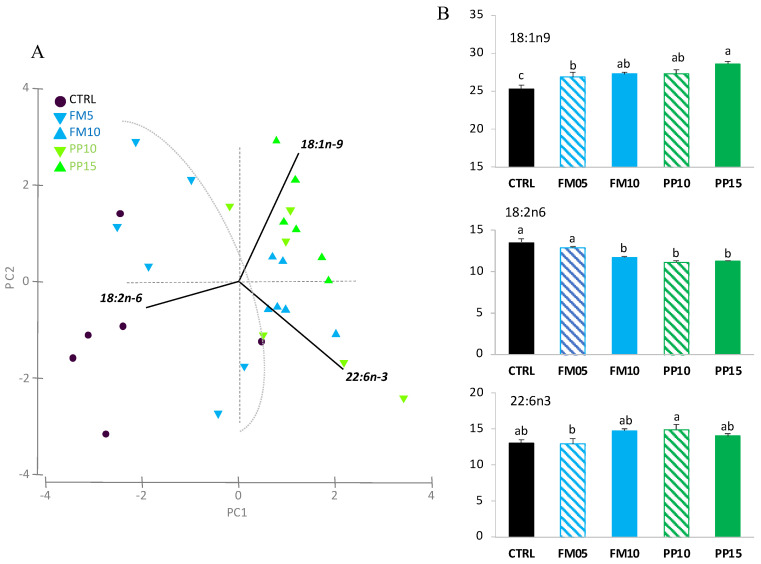
Muscle FA content in fish fed with the control (CTRL, black) or TM-based diets replacing fish meal (blue) or replacing plant meal (green). (**A**) Principal component analysis (PCA) plot based on FA composition (% TFA); (**B**) muscle content of FA significantly correlated with PC1: DHA (22:6n-3), OA (18:1n-9) and LA (18:2n-6); and PC2: 18:1n-9 and 22:6n-3. Data are expressed as mean ± SEM (*n* = 6). Results of two-way ANOVA are presented in the square. Different letters above bars indicate significant different among dietary treatments (Tukey test; *p* < 0.05). FM diets are in blue and PP diets in green. The lowest TM inclusion within each dietary group is indicated with dashed lines.

**Table 1 animals-14-00595-t001:** Formulation and chemical composition (% dw) of Senegalese sole experimental diets (CTRL, FM5, FM10, PP10, and PP15) and *T. molitor* (TM) meal.

		Diets				
	TM	CTRL	FM5	FM10	PP10	PP15
** *Ingredients (% dw)* **						
*Fish meal LT94* ^1^	-	30.00	26.50	23.20	30.00	30.00
*Squid meal* ^2^	-	5.00	5.00	5.00	5.00	5.00
*CPSP90* ^3^	-	5.00	5.00	5.00	5.00	5.00
*Krill meal* ^4^	-	1.00	1.00	1.00	1.00	1.00
*Wheat gluten* ^5^	-	8.70	8.70	8.70	6.80	5.90
*Soybean protein concentrate* ^6^	-	12.00	12.00	12.00	9.90	8.90
*Pea protein concentrate* ^7^	-	9.00	9.00	9.00	7.10	6.20
*Wheat meal* ^8^	-	12.10	12.00	11.60	11.40	10.20
*Tenebrio molitor meal* ^9^	-	0.00	5.00	10.00	10.00	15.00
*Fish oil* ^10^	-	5.40	5.40	5.40	5.00	4.00
*Soybean oil* ^11^	-	3.00	1.60	0.30	0.00	0.00
*Soybean lecithin* ^12^	-	1.00	1.00	1.00	1.00	1.00
*Methionine* ^13^	-	0.50	0.50	0.50	0.50	0.50
*Lysine* ^14^	-	1.20	1.20	1.20	1.20	1.20
*Betaine* ^15^	-	1.00	1.00	1.00	1.00	1.00
*Choline chloride* ^16^	-	0.50	0.50	0.50	0.50	0.50
*Digestive system improver* ^17^	-	0.50	0.50	0.50	0.50	0.50
*Vitamin and mineral premix* ^18^	-	2.00	2.00	2.00	2.00	2.00
*Vitamin C* ^19^	-	0.10	0.10	0.10	0.10	0.10
*Guar gum* ^20^	-	2.00	2.00	2.00	2.00	2.00
** *Proximate composition (% dw)* **						
*Moisture*	7.78	6.49	6.16	6.18	7.29	6.50
*Ash*	2.92	8.07	7.68	7.37	8.03	8.17
*Protein*	43.77	54.03	53.81	53.68	52.95	52.73
*Lipid*	35.05	15.59	15.46	16.51	16.16	17.11

TM: *Tenebrio molitor*; CTRL: control; FM5 and FM10: 5% and 10% of marine animal ingredients replaced by TM meal; PP10 and PP15: 10 and 15% of plant ingredients replaced by TM meal. ^1^ 69.4% crude protein, 12.3% crude lipid (Norsildemel, Bergen, Norway). ^2, 3, 4^ Purchased from Bacarel (UK). CPSP90 is enzymatically pre-digested fishmeal. ^5^ 78% crude protein (Lorca Nutrición Animal SA, Murcia, Spain). ^6^ 50% crude protein, 8% crude lipid (LorcaNutrition, Spain). ^7^ 85% crude protein, 1.5% crude lipid (Emilio Peña SA, Spain). ^8^ Local provider (Almería, Spain). ^9^ Beetle Genius S.L. (Brussels, Belgium). ^10^ AF117DHA (Afamsa, Spain). ^11^ Soybean oil (Aceites el Niño, Spain). ^12^ P700IP (Lecico, DE). ^13, 14, 15, 16^ Lorca Nutrición Animal SA (Murcia, Spain). ^17^ LB_GutHealth_, LifeBioencapsulation S.L. (Almería, Spain) ^18^ Lifebioencapsulation SL (Almería, Spain). Vitamins (mg kg-1): vitamin A (retinyl acetate), 2,000,000 UI; vitamin D3 (DL-cholecalciferol), 200,000 UI; vitamin E (Lutavit E50), 10,000 mg; vitamin K3 (menadione sodium bisulphite), 2500 mg; vitamin B1(thiamine hydrochloride), 3000 mg; vitamin B2 (riboflavin), 3000 mg; calcium pantothenate, 10,000 mg; nicotinic acid, 20,000 mg; vitamin B6 (pyridoxine hydrochloride), 2000 mg; vitamin B9 (folic acid), 1500 mg; vitamin B12 (cyanocobalamin), 10 mg vitamin H (biotin), 300 mg; inositol, 50,000 mg; betaine (Betafin S1), 50,000 mg. Minerals (mg kg-1): Co (cobalt carbonate), 65 mg; Cu (cupric sulphate), 900 mg; Fe (iron sulphate), 600 mg; I (potassium iodide), 50 mg; Mn (manganese oxide), 960 mg; Se (sodium selenite), 1 mg; Zn (zinc sulphate) 750 mg; Ca (calcium carbonate), 18.6%; (186,000 mg); KCl, 2.41%; (24,100 mg); NaCl, 4.0% (40,000 mg). ^19^ TECNOVIT, Spain. ^20^ EPSA, Spain.

**Table 2 animals-14-00595-t002:** Lipid classes (% dw) and fatty acid (% total FA) composition of Senegalese sole experimental diets (CTRL, FM5, FM10, PP10, PP15) and *T. molitor* (TM) meal.

		Diets				
	TM	CTRL	FM5	FM10	PP10	PP15
** *Lipid classes (%TL)* **						
*Lyso-phosphatidylcholine*	0.29	0.84	1.12	1.08	1.25	1.20
*Sphingomyelin*	0.41	0.43	0.26	0.50	0.39	0.65
*Phosphatidylcholine*	1.94	4.97	5.71	5.26	5.44	5.13
*Phosphatidylserine*	0.36	1.52	2.57	2.51	2.07	1.91
*Phosphatidylinositol*	0.38	1.93	1.35	1.62	1.49	1.83
*Phosphatidylethanolamine*	2.54	2.11	2.90	2.51	2.19	2.57
*Diacylglycerol*	3.01	3.19	3.23	3.85	3.48	3.11
*Sterols*	7.36	9.65	10.94	10.20	10.37	9.70
*Free fatty acids*	34.58	13.53	17.29	19.16	19.12	20.09
*Triacylglycerol*	43.45	47.08	42.77	41.90	43.24	42.59
*Sterol esters*	4.10	6.22	6.19	5.19	5.65	5.61
** *Fatty acids (% TFA)* **						
*14:0*	4.18	1.69	1.92	2.16	2.31	2.63
*16:0, PA*	17.98	17.91	18.40	19.13	19.33	19.48
*18:0*	2.55	5.19	4.94	4.81	4.73	4.45
*Total Saturated FA*	24.72	26.19	26.45	27.25	27.44	27.56
*16:1n-7*	2.82	3.41	3.56	3.74	3.93	3.89
*18:1n-9, OA*	51.75	17.96	22.46	26.90	26.64	30.44
*18:1n-7*	0.10	2.37	2.08	1.87	1.90	1.66
*20:1n-9*	0.10	1.52	1.39	1.29	1.37	1.18
*22:1n-11*	nd	0.96	0.89	0.79	0.93	0.80
*Total Monounsaturated FA*	55.86	27.39	31.75	36.02	36.33	39.48
*18:2n-6, LA*	18.00	19.54	16.65	13.89	12.13	12.89
*20:4n-6, ARA*	0.00	1.10	1.06	1.01	1.00	0.83
*22:5n-6*	nd	0.79	0.76	0.73	0.74	0.59
*Total n-6 Polyunsaturated FA*	18.00	22.05	19.13	16.25	14.41	14.82
*18:3n-3, LNA*	0.29	2.42	1.85	1.30	1.13	0.99
*18:4n-3*	nd	0.77	0.71	0.63	0.71	0.61
*20:5n-3, EPA*	nd	5.73	5.38	4.81	5.24	4.50
*22:5n-3, DPA*	nd	0.91	0.99	0.94	0.98	0.82
*22:6n-3, DHA*	nd	13.05	12.37	11.62	12.37	10.01
*Total n-3 Polyunsaturated FA*	0.29	23.61	22.00	19.94	21.12	17.49

TM: *Tenebrio molitor*; CTRL: control; FM5 and FM10: 5% and 10% of marine animal ingredients replaced by TM meal; PP10 and PP15: 10 and 15% of plant ingredients replaced by TM meal. nd: not detected.

**Table 3 animals-14-00595-t003:** Growth performance and nutrient utilization parameters of Senegalese sole fed with the experimental diets during the 98-day feeding trial.

	CTRL	FM5	FM10	PP10	PP15	*p*-Value
Initial body weight (IBW, g)	219.2 ± 5.3	213.7 ± 4.7	215.1 ± 5.2	214.2 ± 5.3	210.8 ± 4.6	*0.979*
Final body weight (FBW, g)	288.6 ± 6.5	284.5 ± 5.8	288.6 ± 6.1	283.8 ± 6.2	288.3 ± 5.5	*0.887*
Specific Growth Rate (SGR, % d^−1^)	0.30 ± 0.01 ^b^	0.32 ± 0.01 ^ab^	0.33 ± 0.01 ^ab^	0.31 ± 0.01 ^b^	0.35 ± 0.01 ^a^	*0.012*
Feed Conversion Ratio (FCR)	1.76 ± 0.20	1.58 ± 0.14	1.61 ± 0.02	1.71 ± 0.17	1.42 ± 0.23	*0.426*
Protein efficiency ratio (PER)	1.00 ± 0.08	1.11 ± 0.08	1.09 ± 0.08	1.04 ± 0.08	1.26 ± 0.08	*0.299*

Dietary treatment codes are CTRL: control; FM5 and FM10: 5% and 10% of marine-derived ingredients replaced by TM meal; PP10 and PP15: 10 and 15 % of plant ingredients replaced by TM meal. Values are mean ± SEM. Values in the same row with different superscript letter indicate significant differences among dietary treatments (*p* < 0.05).

**Table 4 animals-14-00595-t004:** Fatty acid composition (%TFA) in liver of Senegalese sole fed with experimental diets (mean ± SEM, *n* = 6). Values in the same row with different superscript letter indicate significant differences among dietary treatments (*p* < 0.05).

	CTRL	FM5	FM10	PP10	PP15	*p* (Diet)
14:0	3.14 ± 0.21	3.46 ± 0.19	3.42 ± 0.07	3.40 ± 0.16	3.69 ± 0.09	*0.122*
15:0	0.45 ± 0.02	0.38 ± 0.03	0.41 ± 0.03	0.46 ± 0.03	0.40 ± 0.01	*0.092*
16:0	17.29 ± 0.29	17.44 ± 0.32	16.88 ± 0.31	16.69 ± 0.30	16.89 ± 0.15	*0.351*
18:0	3.93 ± 0.38	3.92 ± 0.27	3.72 ± 0.37	3.39 ± 0.24	3.25 ± 0.19	*0.347*
20:0	0.20 ± 0.02	0.19 ± 0.01	0.19 ± 0.01	0.19 ± 0.01	0.19 ± 0.00	*0.974*
22:0	0.18 ± 0.01	0.16 ± 0.01	0.14 ± 0.01	0.14 ± 0.01	0.16 ± 0.01	*0.095*
24:0	0.10 ± 0.02	0.15 ± 0.04	0.11 ± 0.02	0.13 ± 0.02	0.11 ± 0.03	*0.640*
Total Saturated FAs	25.29 ± 0.75	25.70 ± 0.46	24.89 ± 0.56	24.41 ± 0.55	24.69 ± 0.32	*0.396*
16:1n-9	0.78 ± 0.03 ^b^	0.90 ± 0.05 ^ab^	1.02 ± 0.07 ^a^	1.09 ± 0.05 ^a^	1.05 ± 0.03 ^a^	*0.002*
16:1n-7	4.94 ± 0.18	5.54 ± 0.33	5.23 ± 0.12	5.33 ± 0.17	5.48 ± 0.11	*0.306*
18:1n-9	28.09 ± 0.37 ^c^	31.44 ± 1.11 ^b^	32.28 ± 0.54 ^ab^	31.18 ± 0.80 ^b^	34.24 ± 0.59 ^a^	*0.000*
18:1n-7	3.34 ± 0.06	3.19 ± 0.09	3.11 ± 0.05	3.19 ± 0.08	3.01 ± 0.10	*0.103*
20:1n-11	0.15 ± 0.01	0.13 ± 0.01	0.15 ± 0.00	0.15 ± 0.00	0.15 ± 0.01	*0.069*
20:1n-9	1.38 ± 0.06	1.44 ± 0.05	1.44 ± 0.05	1.435 ± 0.07	1.68 ± 0.19	*0.172*
20:1n-7	0.20 ± 0.01	0.20 ± 0.00	0.19 ± 0.01	0.19 ± 0.01	0.18 ± 0.01	*0.566*
22:1n-11	0.51 ± 0.04	0.46 ± 0.03	0.49 ± 0.03	0.54 ± 0.02	0.54 ± 0.04	*0.391*
22:1n-9cis	0.27 ± 0.01	0.27 ± 0.02	0.27 ± 0.01	0.28 ± 0.01	0.27 ± 0.02	*0.986*
24:1n-9	0.31 ± 0.06	0.33 ± 0.03	0.34 ± 0.03	0.35 ± 0.02	0.26 ± 0.06	*0.658*
Total Monounsaturated FAs	40.00 ± 0.56 ^c^	43.92 ± 1.37 ^ab^	44.52 ± 0.48 ^ab^	43.74 ± 0.87 ^b^	46.85 ± 0.47 ^a^	*0.000*
18:2n-6	15.47 ± 0.57 ^a^	13.16 ± 0.76 ^b^	12.57 ± 0.73 ^b^	12.26 ± 0.66 ^b^	12.12 ± 0.13 ^b^	*0.001*
18:3n-6	0.10 ± 0.01	0.10 ± 0.01	0.08 ± 0.00	0.10 ± 0.01	0.09 ± 0.00	*0.252*
20:2n-6	1.18 ± 0.065 ^ab^	1.21 ± 0.06 ^a^	1.03 ± 0.03 ^ab^	0.94 ± 0.04 ^b^	0.94 ± 0.02 ^b^	*0.000*
20:3n-6	0.16 ± 0.01	0.15 ± 0.01	0.16 ± 0.01	0.16 ± 0.01	0.14 ± 0.01	*0.480*
20:4n-6	0.65 ± 0.07	0.66 ± 0.09	0.62 ± 0.06	0.76 ± 0.10	0.61 ± 0.05	*0.698*
22:4n-6	0.27 ± 0.01	0.26 ± 0.03	0.25 ± 0.01	0.26 ± 0.02	0.21 ± 0.01	*0.269*
22:5n-6	0.74 ± 0.03	0.70 ± 0.07	0.71 ± 0.02	0.74 ± 0.05	0.61 ± 0.02	*0.226*
Total n-6 Polyunsaturated FAs	18.57 ± 0.57 ^a^	16.24 ± 1.00 ^ab^	15.42 ± 0.81 ^b^	15.22 ± 0.72 ^b^	14.71 ± 0.14 ^b^	*0.001*
18:3n-3	1.39 ± 0.06 ^a^	1.09 ± 0.05 ^b^	1.03 ± 0.07 ^b^	1.01 ± 0.08 ^b^	0.91 ± 0.05 ^b^	*0.000*
18:4n-3	0.24 ± 0.02 ^ab^	0.18 ± 0.01 ^b^	0.22 ± 0.02 ^ab^	0.25 ± 0.02 ^a^	0.21 ± 0.01 ^ab^	*0.035*
20:3n-3	0.56 ± 0.021 ^a^	0.55 ± 0.02 ^a^	0.45 ± 0.01 ^b^	0.41 ± 0.02 ^b^	0.39 ± 0.02 ^b^	*0.000*
20:4n-3	0.35 ± 0.02 ^ab^	0.29 ± 0.01 ^b^	0.33 ± 0.02 ^ab^	0.36 ± 0.03 ^a^	0.31 ± 0.01 ^ab^	*0.034*
20:5n-3	0.36 ± 0.05 ^ab^	0.25 ± 0.03 ^b^	0.30 ± 0.03 ^ab^	0.40 ± 0.06 ^a^	0.31 ± 0.02 ^ab^	*0.024*
22:5n-3	2.34 ± 0.31	1.83 ± 0.15	2.24 ± 0.06	2.34 ± 0.32	1.96 ± 0.12	*0.260*
22:6n-3	10.13 ± 0.41	9.13 ± 1.00	9.73 ± 0.29	11.05 ± 0.58	8.91 ± 0.33	*0.140*
Total n-3 Polyunsaturated FA	15.41 ± 0.80	13.33 ± 1.16	14.32 ± 0.28	15.83 ± 0.89	13.00 ± 0.48	*0.061*
n-3PUFA/n-6PUFA	0.83 ± 0.04	0.82 ± 0.06	0.94 ± 0.05	1.05 ± 0.07	0.88 ± 0.03	*0.048*

**Table 5 animals-14-00595-t005:** Fatty acid composition (%TFA) in muscle of Senegalese sole fed with experimental diets (mean ± SEM, *n* = 6). Values in the same row with different superscript letter indicate significant differences among dietary treatments (*p* < 0.05).

	CTRL	FM5	FM10	PP10	PP15	*p* (Diet)
14:0	2.30 ± 0.13	2.36 ± 0.12	1.86 ± 0.16	1.99 ± 0.21	2.16 ± 0.11	*0.190*
15:0	0.48 ± 0.02 ^a^	0.47 ± 0.02 ^ab^	0.40 ± 0.02 ^b^	0.40 ± 0.02 ^ab^	0.40 ± 0.01 ^b^	*0.009*
16:0	17.79 ± 0.25	18.11 ± 0.12	17.97 ± 0.21	18.08 ± 0.17	18.43 ± 0.10	*0.223*
18:0	3.54 ± 0.08 ^b^	3.63 ± 0.17 ^ab^	4.10 ± 0.13 ^a^	3.82 ± 0.15 ^ab^	3.90 ± 0.03 ^ab^	*0.035*
20:0	0.25 ± 0.01	0.25 ± 0.01	0.25 ± 0.01	0.26 ± 0.01	0.26 ± 0.01	*0.457*
22:0	0.19 ± 0.01 ^b^	0.19 ± 0.00 ^ab^	0.18 ± 0.01 ^ab^	0.17 ± 0.01 ^b^	0.18 ± 0.00 ^ab^	*0.035*
24:0	0.08 ± 0.03	0.10 ± 0.02	0.12 ± 0.02	0.10 ± 0.01	0.07 ± 0.02	*0.657*
Total Saturated FAs	24.65 ± 0.41	25.11 ± 0.19	24.88 ± 0.38	24.83 ± 0.34	25.39 ± 0.19	*0.568*
16:1n-9	0.47 ± 0.01 ^b^	0.54 ± 0.01 ^a^	0.55 ± 0.02 ^a^	0.56 ± 0.01 ^a^	0.58 ± 0.01 ^a^	*0.000*
16:1n-7	4.79 ± 0.12	4.76 ± 0.19	4.18 ± 0.12	4.40 ± 0.21	4.35 ± 0.07	*0.031*
18:1n-9	25.33 ± 0.47 ^c^	26.88 ± 0.62 ^b^	27.29 ± 0.22 ^ab^	27.31 ± 0.48 ^ab^	28.62 ± 0.30 ^a^	*0.000*
18:1n-7	2.80 ± 0.12 ^a^	2.70 ± 0.19 ^ab^	2.56 ± 0.12 ^bc^	2.58 ± 0.21 ^bc^	2.43 ± 0.07 ^c^	*0.000*
20:1n-11	0.21 ± 0.01 ^a^	0.20 ± 0.01 ^ab^	0.19 ± 0.01 ^ab^	0.201 ± 0.00 ^ab^	0.19 ± 0.01 ^b^	*0.023*
20:1n-9	1.51 ± 0.05	1.45 ± 0.04	1.40 ± 0.02	1.43 ± 0.04	1.36 ± 0.02	*0.084*
20:1n-7	0.16 ± 0.01	0.16 ± 0.01	0.15 ± 0.01	0.15 ± 0.01	0.15 ± 0.01	*0.860*
22:1n-11	0.76 ± 0.03	0.71 ± 0.02 ^ab^	0.62 ± 0.02 ^b^	0.69 ± 0.03 ^ab^	0.64 ± 0.01 ^b^	*0.002*
22:1n-9cis	0.29 ± 0.01	0.29 ± 0.01	0.26 ± 0.03	0.28 ± 0.01	0.27 ± 0.02	*0.157*
24:1n-9	0.40 ± 0.01	0.22 ± 0.10	0.41 ± 0.03	0.43 ± 0.01	0.41 ± 0.02	*0.046*
Total Monounsaturated FAs	36.72 ± 0.68	37.89 ± 0.92	37.61 ± 0.35	38.05 ± 0.67	39.01 ± 0.43	*0.100*
18:2n-6	13.50 ± 0.52 ^a^	12.86 ± 0.19 ^a^	11.69 ± 0.15 ^b^	11.15 ± 0.19 ^b^	11.25 ± 0.06 ^b^	*0.000*
18:3n-6	0.14 ± 0.01	0.14 ± 0.01	0.14 ± 0.01	0.13 ± 0.01	0.12 ± 0.01	*0.117*
20:2n-6	0.66 ± 0.01 ^abc^	0.68 ± 0.01 ^a^	0.67 ± 0.00 ^ab^	0.62 ± 0.01 ^bc^	0.62 ± 0.01 ^c^	*0.001*
20:3n-6	0.17 ± 0.01	0.16 ± 0.01	0.17 ± 0.01	0.18 ± 0.01	0.16 ± 0.01	*0.341*
20:4n-6	0.82 ± 0.04 ^b^	0.83 ± 0.06 ^b^	1.01 ± 0.03 ^a^	0.98 ± 0.06 ^ab^	0.96 ± 0.03 ^ab^	*0.008*
22:4n-6	0.28 ± 0.01 ^ab^	0.29 ± 0.01 ^b^	0.30 ± 0.00 ^a^	0.30 ± 0.01 ^ab^	0.27 ± 0.01 ^b^	*0.009*
22:5n-6	0.61 ± 0.02	0.63 ± 0.03	0.70 ± 0.02	0.69 ± 0.04	0.66 ± 0.01	*0.061*
Total n-6 Polyunsaturated FAs	16.19 ± 0.50 ^a^	15.60 ± 0.16 ^b^	14.68 ± 0.15 ^c^	14.05 ± 0.13 ^c^	14.03 ± 0.06 ^c^	*0.000*
18:3n-3	1.85 ± 0.08 ^a^	1.60 ± 0.05 ^b^	1.35 ± 0.03 ^c^	1.32 ± 0.06 ^c^	1.20 ± 0.03 ^c^	*0.000*
18:4n-3	0.64 ± 0.04 ^a^	0.57 ± 0.03 ^ab^	0.49 ± 0.02 ^b^	0.52 ± 0.03 ^b^	0.46 ± 0.01 ^b^	*0.001*
20:3n-3	0.36 ± 0.01 ^a^	0.35 ± 0.01 ^a^	0.32 ± 0.01 ^ab^	0.29 ± 0.01 ^b^	0.28 ± 0.01 ^b^	*0.000*
20:4n-3	0.44 ± 0.02 ^a^	0.41 ± 0.01 ^ab^	0.39 ± 0.00 ^b^	0.39 ± 0.01 ^b^	0.37 ± 0.01 ^b^	*0.001*
20:5n-3	1.69 ± 0.07 ^a^	1.43 ± 0.03 ^ab^	1.46 ± 0.04 ^b^	1.57 ± 0.06 ^b^	1.40 ± 0.04 ^b^	*0.001*
22:5n-3	3.76 ± 0.08 ^a^	3.50 ± 0.12 ^ab^	3.54 ± 0.05 ^ab^	3.54 ± 0.12 ^ab^	3.28 ± 0.09 ^b^	*0.043*
22:6n-3	13.04 ± 0.48 ^ab^	12.93 ± 0.70 ^b^	14.77 ± 0.27 ^ab^	14.92 ± 0.69 ^a^	14.05 ± 0.34 ^ab^	*0.012*
Total n-3 Polyunsaturated FAs	21.79 ± 0.44	20.80 ± 0.73	22.32 ± 0.22	22.56 ± 0.63	21.07 ± 0.44	*0.045*
n-3PUFA/n-6PUFA	1.35 ± 0.05 ^bc^	1.33 ± 0.040 ^c^	1.52 ± 0.01 ^a^	1.61 ± 0.05 ^a^	1.50 ± 0.03 ^ab^	*0.000*

## Data Availability

Data will be made available upon any reasonable request to the corresponding author.

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
