# Peer review of "Assessment of Full-Fat Tenebrio molitor as Feed Ingredient for Solea senegalensis: Effects on Growth Performance and Lipid Profile"

_animals, 2024, doi:10.3390/ani14040595_

Round 1
Reviewer 1 Report
Comments and Suggestions for Authors
Studies that examine the effects of different protein sources and feed formulations on the growth, production and well-being of farmed fish are appropriate for the journal.
There are shortcomings in the way in which the authors have organized and presented their material, and with the way they have structured the manuscript.
There were shortcomings with the trial, there are problems with some of the reporting and some of the data analysis seems to be flawed.
In a growth trial it should be ensured that the fish increase by at least 2-to-3fold in body mass, but this was not the case in the study reported in this manuscript. The fish increased in body mass from ca 215 g to ca 285-290 g over 98-days (Table 3)
There were three tanks of fish used for each feed treatment, meaning that n = 3 for the purposes of data presentation and statistical analysis. In the figure legends, table headings and footnotes the values for n are sometimes given as n = 6, which seems to indicate that there were problems with the analyses relating to pseudo-replication.
The authors took samples of the fish liver and muscle for the analysis of proximate chemical composition and FA analyses (lines 197-199). However, they only analyzed protein and lipid in the muscle, and they present the data as % dry mass (Figure 2). Data should be given in terms of % wet mass, because of the risk of incorrect interpretation when reporting is as % dry mass.
Note: On line 198 it is stated that ‘muscle’ was sampled, but in the legend to figure 2 the term ‘fillet’ is used.
The authors have collected quite a lot of information, so it is unfortunate that there are problems with the sampling, analyses, and reporting.
There are shortcomings with the technical presentation.
Make sure that all works cited in the text are used correctly, that they are in the reference list, that the presentation is consistent and that correct information is given in the reference list.
Define and explain all acronyms and abbreviations on first mention in the text, make sure that the presentation is correct and be consistent in the way that they are used.
On first mention of a species in the text give both the common (trivial) and formal names, and make sure that the presentation is correct and consistent.
Make sure that symbols, sub- and super-scripts, upper- and lower-case are presented correctly, and that there is correct and consistent use of italics, brackets, and punctuation etc.
The authors do not have complete command of English. Linguistic changes and corrections will be required to improve the language presentation.
It is unfortunate that there are weaknesses in the presentation that have resulted in a text that is sometimes incomplete or imprecise, and a manuscript that has several deficiencies.
The Abstract has a poor structure, being disjointed, and with a text that is lacking in cohesion and continuity. There is insufficient information about the study design. There are several imprecise and confusing statements, and the information given is selective, and incomplete. For example, what were the nutrient compositions (protein and lipid) of the feeds and what was the feeding routine? What was the size of the fish, what was tank size and stocking density? What was the treatment replication and for how long did the trial last? What was the sampling protocol and which metrics were assessed?
The presentation of the findings is too imprecise because there is no quantitative (numerical) information given about the responses of the fish to the different treatments.
The Abstract will need to be completely re-worked and re-written.
The Introduction contains some useful background information, but the presentation is disjointed and staccato, and there are imprecise statements. This means that the text lacks cohesion and continuity, and that the content is likely to mislead and confuse readers.
For example, the Introduction opens with some general statements about protein sources, but the inclusion of source [4] is incorrect because this covers plant oils, and it is not an overview of the topic.
Some re-structuring and re-writing of the Introduction will be required to correct the mistakes and improve the presentation.
The amount of information given about the M & Ms is variable, with some protocols being described in sufficient detail, but insufficient information being given about others. There are problems with the presentation of the M & Ms because not all information is given sequentially, there are several imprecise or confusing statements, and there are a few mistakes.
The authors give some details about the TM in section 2.1 and in tables 1 and 2, but the information is imprecise and confusing. Details about the proximate chemical and FA compositions are presented before any information about the analyses (given in section 2.6). In the text (line 116), the authors mention OA, LA, and PA but in table 2 give the FA shorthand formula. This means that readers need to return to the Ìntroduction (lines 81-83) to see the connection.
Similarly, the information about the proximate chemical and FA compositions of the feeds (section 2.2) is given before any details of the analyses that were used to collect the data (section 2.6).
How were the treatment groups established, and data for initial size obtained? Was each fish weighed, and the tag-number and tank placement noted immediately prior to the start of the growth trial (section 2.3)?
The description of the feeding protocol, making of feed supply adjustments and recording of feed intake is incomplete, imprecise, and unclear. The text will need to be re-written to give more details and make the presentation clearer.
At the end of the trial did the sampling involve taking 2 fish from each tank per treatment?
How were the fish dissected to obtain the liver and muscle samples? Samples were used for FA analysis; were they stored in a nitrogen atmosphere?
The sampling and analyses were insufficient to collect good, comprehensive data relating to nutrient utilization. Initial and final samples of fish should have been taken and analyzed to obtain the information needed to make assessments of nutrient deposition and nutrient retention.
In section 2.7 the authors state that they examined the effects of ‘gender’ on muscle and liver composition. However, earlier in the M & Ms they make no mention of having determined the sex of the fish they sampled, and the numbers of fish sampled from each tank and feed treatment would seem to limit the usefulness of such an analysis.
Given the problems with the way in which the authors present the work, including incomplete, imprecise, and confusing information about the M & Ms, and concerns about the some of the data analyses, it would be premature to provide detailed comments about the Results and Discussion sections of the manuscript.
Comments on the Quality of English LanguageSome corrections and structural improvements required
Author Response
REVIEWER #1:
General comments:
Reviewer: In a growth trial it should be ensured that the fish increase by at least 2-to-3fold in body mass, but this was not the case in the study reported in this manuscript. The fish increased in body mass from ca 215 g to ca 285-290 g over 98-days (Table 3)
Reply: This is normal when you are experimenting with juvenile fish. However, in the present work we have evaluated animals close to commercial weight, and with these animals we will need much time for the fish to increase their body mass 2 to 3 times. We believe that the duration of the experiment and the increase in weight are sufficient to evaluate growth and the effect on the lipid composition of the tissues.
Below I give some published work with similar weight gain during the experimental period.
- Luis Molina-Roque, André Bárany, María Isabel Sáez, Francisco Javier Alarcón, Silvana Teresa Tapia, Juan Fuentes, Juan Miguel Mancera, Erick Perera, Juan Antonio Martos-Sitcha. Biotechnological treatment of microalgae enhances growth performance, hepatic carbohydrate metabolism and intestinal physiology in gilthead seabream (Sparus aurata) juveniles close to commercial size. (2022). Aquaculture Reports, Volume 25, 101248, ISSN 2352-5134, https://doi.org/10.1016/j.aqrep.2022.101248.
- Saez, María & Galafat, Alba & Suarez, Md & Chaves-Pozo, Elena & Arizcun, Marta & Ayala, M. & Alarcón, Fco & Martínez Moya, Tomás. (2023). Effects of raw and hydrolysed Nannochloropsis gaditana biomass included at low level in finishing diets for gilthead seabream (Sparus aurata) on fillet quality and shelf life. Journal of Applied Phycology. 35. 1-19. 10.1007/s10811-023-02957-6.
- Ayala, M.D.; Chaves-Pozo, E.; Sáez, M.I.; Galafat, A.; Alarcón, F.J.; Martínez, T.F.; Arizcun, M. Effect on Muscle Cellularity of Diet Supplementation with Nannochloropsis gaditana Microalgae in the Final Fattening Phase of Gilthead Seabream Culture up to Commercial Size. Fishes 2023, 8, 532. https://doi.org/10.3390/fishes8110532
Reviewer: There were three tanks of fish used for each feed treatment, meaning that n = 3 for the purposes of data presentation and statistical analysis. In the figure legends, table headings and footnotes the values for n are sometimes given as n = 6, which seems to indicate that there were problems with the analyses relating to pseudo-replication.
Reply: As explained on page 6, line197 of the submitted manuscript (SM), “In the last sampling, thirty fish (six fish by diet) were sacrificed “. The total samples of liver and muscle analyses were n=6, 2 fish per tank replicate, which means 6 fish per diet. We have clarified in the methodology (p.7, L.220 of the reviewed manuscript, RM).
Reviewer: The authors took samples of the fish liver and muscle for the analysis of proximate chemical composition and FA analyses (lines 197-199). However, they only analyzed protein and lipid in the muscle, and they present the data as % dry mass (Figure 2). Data should be given in terms of % wet mass, because of the risk of incorrect interpretation when reporting is as % dry mass.
Reply: We do not agree that data expressed as % dry mass can lead to incorrect interpretation, in fact we believe that it is more accurate as it eliminates the effect of moisture. In any case, we can change the way the data is expressed if the reviewer thinks it is appropriate.
Reviewer: On line 198 it is stated that ‘muscle’ was sampled, but in the legend to figure 2 the term ‘fillet’ is used.
Reply: We have homogenised the term used and now only use the term "muscle".
Reviewer: Make sure that all works cited in the text are used correctly, that they are in the reference list, that the presentation is consistent and that correct information is given in the reference list.
Define and explain all acronyms and abbreviations on first mention in the text, make sure that the presentation is correct and be consistent in the way that they are used.
On first mention of a species in the text give both the common (trivial) and formal names, and make sure that the presentation is correct and consistent.
Make sure that symbols, sub- and super-scripts, upper- and lower-case are presented correctly, and that there is correct and consistent use of italics, brackets, and punctuation etc.
The authors do not have complete command of English. Linguistic changes and corrections will be required to improve the language presentation.
Reply: We have checked it and fixed all the mistakes we found.
Reviewer: The Abstract has a poor structure, being disjointed, and with a text that is lacking in cohesion and continuity. There is insufficient information about the study design. There are several imprecise and confusing statements, and the information given is selective, and incomplete. For example, what were the nutrient compositions (protein and lipid) of the feeds and what was the feeding routine? What was the size of the fish, what was tank size and stocking density? What was the treatment replication and for how long did the trial last? What was the sampling protocol and which metrics were assessed?
The presentation of the findings is too imprecise because there is no quantitative (numerical) information given about the responses of the fish to the different treatments.
The Abstract will need to be completely re-worked and re-written.
Reply: Abstract has been completely rewritten and all information requested by the reviewer has been added.
Reviewer: The Introduction contains some useful background information, but the presentation is disjointed and staccato, and there are imprecise statements. This means that the text lacks cohesion and continuity, and that the content is likely to mislead and confuse readers.
For example, the Introduction opens with some general statements about protein sources, but the inclusion of source [4] is incorrect because this covers plant oils, and it is not an overview of the topic.
Some re-structuring and re-writing of the Introduction will be required to correct the mistakes and improve the presentation.
Reply: Introduction has been corrected and completely rewritten. All sources have been reviewed.
Reviewer: The authors give some details about the TM in section 2.1 and in tables 1 and 2, but the information is imprecise and confusing.
Reply: We consider that the information about TM is clear and sufficient. However, we have rewritten section 2.1. and added some data.
Reviewer: Details about the proximate chemical and FA compositions are presented before any information about the analyses (given in section 2.6). In the text (line 116), the authors mention OA, LA, and PA but in table 2 give the FA shorthand formula. This means that readers need to return to the Introduction (lines 81-83) to see the connection.
Similarly, the information about the proximate chemical and FA compositions of the feeds (section 2.2) is given before any details of the analyses that were used to collect the data (section 2.6).
Reply: It is common practice in many nutrition works to present dietary composition data before describing the biochemical analyses. This is not a problem in our view as the analysis methodology will be detailed below.
Below I give some examples:
- Valeria Iaconisi, Antonio Bonelli, Rita Pupino, Francesco Gai, Giuliana Parisi. Mealworm as dietary protein source for rainbow trout: Body and fillet quality traits. (2018). Aquaculture, Volume 484, Pages 197-204, ISSN 0044-8486, https://doi.org/10.1016/j.aquaculture.2017.11.034.
- Dmitri Fabrikov, Fernando G. Barroso, Mª. José Sánchez-Muros, Mª. Carmen Hidalgo, Gabriel Cardenete, Cristina Tomás-Almenar, Federico Melenchón, Jose Luis Guil-Guerrero. Effect of feeding with insect meal diet on the fatty acid compositions of sea bream (Sparus aurata), tench (Tinca tinca) and rainbow trout (Oncorhynchus mykiss) fillets. (2021). Aquaculture, Volume 545, 2021, 737170, ISSN 0044-8486, https://doi.org/10.1016/j.aquaculture.2021.737170.
- Pedro Borges, Bruno Reis, Telmo J.R. Fernandes, Ângela Palmas, Manuela Castro-Cunha, Françoise Médale, Maria Beatriz P.P. Oliveira, Luísa M.P. Valente, Senegalese sole juveniles can cope with diets devoid of supplemental fish oil while preserving flesh nutritional value (2014) Aquaculture, Volumes 418–419, 2014, Pages 116-125, ISSN 0044-8486, https://doi.org/10.1016/j.aquaculture.2013.10.014.
Reviewer: In the text (line 116), the authors mention OA, LA, and PA but in table 2 give the FA shorthand formula. This means that readers need to return to the Ìntroduction (lines 81-83) to see the connection.
Reply: We have included shorthand formula when mention these FA (p.3, L.124 of the reviewed manuscript, RM).
Reviewer: How were the treatment groups established, and data for initial size obtained? Was each fish weighed, and the tag-number and tank placement noted immediately prior to the start of the growth trial (section 2.3)?
Reply: The methodology has been completed by adding that information (p.7, L.194-203 of the reviewed manuscript, RM).
Reviewer: The description of the feeding protocol, making of feed supply adjustments and recording of feed intake is incomplete, imprecise, and unclear. The text will need to be re-written to give more details and make the presentation clearer.
Reply: This part has been rewritten.
Reviewer: At the end of the trial did the sampling involve taking 2 fish from each tank per treatment?
Reply: At the end all fish were weighted and 6 fish per treatment (2 fish per tank) were sacrificed and dissected.
Reviewer: How were the fish dissected to obtain the liver and muscle samples? Samples were used for FA analysis; were they stored in a nitrogen atmosphere?
Reply: Fish were cold dissected. Samples were immediately frozen at -80°C in liquid nitrogen. These aspects have been clarified in the text (p.7, L.221-222 of the reviewed manuscript, RM).
Reviewer: The sampling and analyses were insufficient to collect good, comprehensive data relating to nutrient utilization. Initial and final samples of fish should have been taken and analyzed to obtain the information needed to make assessments of nutrient deposition and nutrient retention.
Reply: In experiments with fish of this size, where developmental changes are not expected, it is not necessary to compare the results of the final samples with those of the initial samples. In the above-mentioned references, you will find a number of examples of work in the nutrition field in which a sample is not taken at the beginning.
Reviewer: In section 2.7 the authors state that they examined the effects of ‘gender’ on muscle and liver composition. However, earlier in the M & Ms they make no mention of having determined the sex of the fish they sampled, and the numbers of fish sampled from each tank and feed treatment would seem to limit the usefulness of such an analysis.
Reply: I'm sorry, but there is an error in this part of the material and method, the sex of the animals was not analysed. This part has been corrected in the new version of the manuscript.
Reviewer 2 Report
Comments and Suggestions for Authors
The present study evaluated the effects of mealworm meal (Tenebrio molitor) as feed ingredient on growth performance and lipid profile of Senegalese sole. The experimental design was reasonable, and the manuscript was well written. A few points the authors could address in the present manuscript before publishing.
1. For growth, only 0.30-0.35% SGR were observed in the 98-day feeding trial. It looks the growth rate was poor for species. Was it normal for the Senegalese sole?
2. L. 251-253, the statement "Two-way ANOVAs were run to test the effect of diets and gender on muscle fillets and liver protein and lipid composition at the end of trial" is confused. Did the authors use different gender of fish in this study? If yes, how many males and females in each tanks must be given. In addition, no 2-way ANOVA results was found in Tables. If the authors did that, please provide the data and discussion in the manuscript.
Author Response
REVIEWER #2:
Reviewer: For growth, only 0.30-0.35% SGR were observed in the 98-day feeding trial. It looks the growth rate was poor for species. Was it normal for the Senegalese sole?
Reply: The truth is that these growth rates are rather low for this species. There are two points that can explain this:
- Growth during the first days (0-43) was very low (Figure 1), probably due to the animals' adaptation to the experimental conditions. This reduced the average growth throughout the experimental period.
- The fish are close to commercial size, and at this size growth is lower compared to juveniles
Reviewer: L. 251-253, the statement "Two-way ANOVAs were run to test the effect of diets and gender on muscle fillets and liver protein and lipid composition at the end of trial" is confused. Did the authors use different gender of fish in this study? If yes, how many males and females in each tanks must be given. In addition, no 2-way ANOVA results was found in Tables. If the authors did that, please provide the data and discussion in the manuscript.
Reply: I'm sorry, but there is an error in this part of the material and method, the sex of the animals was not analysed. This part has been corrected in the new version of the manuscript.
Reviewer 3 Report
Comments and Suggestions for Authors
It is valuable work; however, I made some remarks for further improvements that can be found in the attached MS.
Your experimental setup—testing two versions of the same treatment—was quite original. However, of course, making it separately in two experiments would certainly be more unequivocal. Even just out of curiosity, I should try to evaluate these data according to that to see the differences.

Author Response
REVIEWER #3:
Reviewer: Your experimental setup—testing two versions of the same treatment—was quite original. However, of course, making it separately in two experiments would certainly be more unequivocal. Even just out of curiosity, I should try to evaluate these data according to that to see the differences.
Reply: We have re-evaluated data separately (plant or fish meal replacement effect) and the results and main conclusions are quite similar. In our opinion, the paper is more readable (fewer figures and tables) and the analysis is more complete when all treatments are evaluated together.
Reviewer: about word “replaced mostly …. meal”. It's a bit unclear in this way. See my remarks later.
Reply: We don't really understand what the reviewer means. We use "mostly" because the main ingredient replaced by TM is fishmeal or plant meal, but to adjust the fat content of the diets, soybean oil is also adjusted (see Table 1).
Reviewer: As I see sex determination was not mentioned till now and there are no concrete results given, nether n number. Provide that or omit this part.
Reply: I'm sorry, but there is an error in this part of the material and method, the sex of the animals was not analysed. We omit this part.
Reviewer: Possible causes of this inflection point should be explained or give some conjectures about it. After all it is the only interesting outcome given by the two-way ANOVA.
Reply: We think that this inflection point is probably due to the adaptation of the animals to the experimental conditions. This point has been clarified in the text (p.9, L.296-298 of the revised manuscript, RM).
Reviewer: Can these values considered to be good ones? Salas-Leiton et al., 2010 published double ones at the same feeding ration.
Reply: The truth is that these growth rates are rather low for this species. There are two points that can explain this:
- Growth during the first days (0-43) was very low (Figure 1), probably due to the animals' adaptation to the experimental conditions. This reduced the average growth throughout the experimental period.
- The fish are close to commercial size, and at this size growth is lower compared to juveniles
Regarding the comparison with the SGR reported for Salas Leiton et al. 2010, our SGR values are around 0.35-0.4, if we do not consider the first period (0-43), and they are not so far from those reported by this author (0.4-0.5).
Reviewer: This part is quite laconic. Something about the replacement of FM and plant protein sources? It may justify your really original experimental design after all.
Reply: We have re-write this section.
Round 2
Reviewer 1 Report
Comments and Suggestions for Authors
The authors have made quite a lot of changes and revisions, but the text is difficult to read because I was not provided with a ‘clean’ copy of the manuscript to evaluate.
However, it was possible to see that the authors had not checked the manuscript with sufficient care prior to making their re-submission. There are several careless mistakes in the text and tables.
Although the authors have made changes to the presentation of the work in the manuscript, they were unable to overcome the problems created by the shortcomings in the study design, sampling, and analyses.
In a growth trial it should be ensured that the fish increase by at least 2-to-3fold in body mass, but this was not the case in the study reported in this manuscript. The fish increased in body mass from ca 215 g to ca 285-290 g over 98-days (Table 3). There may have been several reasons for this seemingly poor growth. The authors fed a ‘restricted ration’ (0.5-1% biomass) (lines 290-294) and this may have imposed a growth limitation on the fish. The growth of the fish was poorest during the early part of the trial (figure 1), and this may indicate that the fish were not adequately acclimatized to the test conditions when the trial started.
The authors give data for lipid and protein of the muscle (figure 1) and liver (line 424) in terms of % dry mass. This is a poor form of presentation because it can result in dubious interpretation and the drawing of incorrect conclusions. Proximate chemical composition data should be given in terms of % wet mass.
There were three tanks of fish used for each feed treatment, meaning that n = 3 for the purposes of data presentation and statistical analysis. In the figure legends, table headings and footnotes the values for n are sometimes given as n = 6, which seems to indicate that there were problems with the analyses relating to pseudo-replication.
Comments on the Quality of English LanguageSome linguistic changes and corrections needed. Quite a lot of careless mistakes in spelling and grammar.
Author Response
REVIEWER #1:
GENERAL COMMENTS:
Reviewer: The authors have made quite a lot of changes and revisions, but the text is difficult to read because I was not provided with a ‘clean’ copy of the manuscript to evaluate.
However, it was possible to see that the authors had not checked the manuscript with sufficient care prior to making their re-submission. There are several careless mistakes in the text and tables.
Although the authors have made changes to the presentation of the work in the manuscript, they were unable to overcome the problems created by the shortcomings in the study design, sampling, and analyses.
Reply: The manuscript has been carefully reviewed, resulting in the correction of all grammatical errors and mistakes in both the text and tables. Two versions of the manuscript have been submitted: a clean copy and a version with the changes and revisions highlighted in order to facilitate the review process.
We have re-analysed and presented the data as suggested by the reviewer. We appreciate the reviewer's feedback and would like to address their concerns regarding the experimental design. In our response to the specific comments below, we provide further clarification and evidence to support our approach.
SPECIFIC COMMENTS:
Reviewer: In a growth trial it should be ensured that the fish increase by at least 2-to-3fold in body mass, but this was not the case in the study reported in this manuscript. The fish increased in body mass from ca 215 g to ca 285-290 g over 98-days (Table 3). There may have been several reasons for this seemingly poor growth. The authors fed a ‘restricted ration’ (0.5-1% biomass) (lines 290-294) and this may have imposed a growth limitation on the fish. The growth of the fish was poorest during the early part of the trial (figure 1), and this may indicate that the fish were not adequately acclimatized to the test conditions when the trial started.
Reply: As we indicated in the first revision, a 2-3-fold increase in fish body mass is normal when experimenting with juvenile fish. However, in this study we evaluated animals that were close to commercial weight. Therefore, it would take a considerable amount of time for the fish to increase their body mass by 2-3 times. Moreover, the main objective of this piece of research in to assess the effect of diet on tissue in fatty acid composition. We believe that the duration of the experiment and the weight gain are enough to evaluate growth and the effect on tissue lipid composition. Below are detailed several published papers with similar objectives, where similar weight gain during the experimental period are reported.
- Luis Molina-Roque, André Bárany, María Isabel Sáez, Francisco Javier Alarcón, Silvana Teresa Tapia, Juan Fuentes, Juan Miguel Mancera, Erick Perera, Juan Antonio Martos-Sitcha. Biotechnological treatment of microalgae enhances growth performance, hepatic carbohydrate metabolism and intestinal physiology in gilthead seabream (Sparus aurata) juveniles close to commercial size. (2022). Aquaculture Reports, Volume 25, 101248, ISSN 2352-5134, https://doi.org/10.1016/j.aqrep.2022.101248.
- Saez, María & Galafat, Alba & Suarez, Md & Chaves-Pozo, Elena & Arizcun, Marta & Ayala, M. & Alarcón, Fco & Martínez Moya, Tomás. (2023). Effects of raw and hydrolysed Nannochloropsis gaditana biomass included at low level in finishing diets for gilthead seabream (Sparus aurata) on fillet quality and shelf life. Journal of Applied Phycology. 35. 1-19. 10.1007/s10811-023-02957-6.
- Ayala, M.D.; Chaves-Pozo, E.; Sáez, M.I.; Galafat, A.; Alarcón, F.J.; Martínez, T.F.; Arizcun, M. Effect on Muscle Cellularity of Diet Supplementation with Nannochloropsis gaditana Microalgae in the Final Fattening Phase of Gilthead Seabream Culture up to Commercial Size. Fishes 2023, 8, 532. https://doi.org/10.3390/fishes8110532
With regards to the growth of the fish, it is acknowledged that acclimatization to the experimental conditions may have caused slower growth at the beginning. However, it is worth noting that growth later recovered and reached values similar to those previously reported by Salas-Leiton et al. (2010) and Sánchez et al (2010). It is also important to mention that the specific growth rate (SGR) values, excluding the first period, are consistent with the values reported by these authors (0.3-0.5), ranging from approximately 0.35-0.4. A comment has been added to the discussion section (page 16, lines 409-411 of the reviewed manuscript (RM)).
In response to the statement that a restricted ration (0.5-1% biomass) was provided, we would like to respectfully express our disagreement. Our methodology was closely monitored and we stand by our findings. As indicated on page 7, lines 209-211 of the reviewed manuscript (RM): “we adjusted the daily feed supply based on the amount of feed remaining in the tanks. This approach allowed the animals to be fed ad libitum, and if no leftovers were observed, the ration was increased”.
Salas-Leiton, E., Anguís, V., Rodríguez-Rúa, A., and Cañavate, J.P. (2010). Stocking homogeneous size groups does not improve growth performance of Senegalese sole (Solea senegalensis, Kaup 1858) juveniles: Individual growth related to fish size. Aquacultural Engineering 43, 108-113.
Sánchez, P., Ambrosio, P.P., and Flos, R. (2010). Stocking density and sex influence individual growth of Senegalese sole (Solea senegalensis). Aquaculture 300, 93-101.
Reviewer: The authors give data for lipid and protein of the muscle (figure 1) and liver (line 424) in terms of % dry mass. This is a poor form of presentation because it can result in dubious interpretation and the drawing of incorrect conclusions. Proximate chemical composition data should be given in terms of % wet mass.
Reply: The data for lipid and protein content in the muscle (figure 1) and liver (page 10, line 325 of the reviewed manuscript (RM)) are now presented as a percentage of wet mass, and have been updated.
Reviewer: There were three tanks of fish used for each feed treatment, meaning that n = 3 for the purposes of data presentation and statistical analysis. In the figure legends, table headings and footnotes the values for n are sometimes given as n = 6, which seems to indicate that there were problems with the analyses relating to pseudo-replication.
Reply: As you suggested, we have reanalysed the data, taking into account n=6, as in the earlier versions.
Round 3
Reviewer 1 Report
Comments and Suggestions for Authors
Based on my verdict following the evaluation of the previous version I am not going to comment on this one